# The use of artificial intelligence tools in cancer detection compared to the traditional diagnostic imaging methods: An overview of the systematic reviews

Helbert Eustáquio Cardoso da Silva👁️*◎, Glaucia Nize Martins Santos◎, André Ferreira Leite‡, Carla Ruffeil Moreira Mesquita‡, Paulo Tadeu de Souza Figueiredo‡, Cristine Miron Stefani👁️◎, Nilce Santos de Melo◎

Faculty of Health Science, Dentistry of Department, Brasilia University, Brasilia, Brazil

◎ These authors contributed equally to this work.
‡ These authors also contributed equally to this work
* helbertcardososilva@gmail.com

**Data Availability Statement:** All relevant data are within the paper and its Supporting Information files.

## Abstract

### Background and purpose

In comparison to conventional medical imaging diagnostic modalities, the aim of this overview article is to analyze the accuracy of the application of Artificial Intelligence (AI) techniques in the identification and diagnosis of malignant tumors in adult patients.

### Data sources

The acronym PIRDs was used and a comprehensive literature search was conducted on PubMed, Cochrane, Scopus, Web of Science, LILACS, Embase, Scielo, EBSCOhost, and grey literature through Proquest, Google Scholar, and JSTOR for systematic reviews of AI as a diagnostic model and/or detection tool for any cancer type in adult patients, compared to the traditional diagnostic radiographic imaging model. There were no limits on publishing status, publication time, or language. For study selection and risk of bias evaluation, pairs of reviewers worked separately.

### Results

In total, 382 records were retrieved in the databases, 364 after removing duplicates, 32 satisfied the full-text reading criterion, and 09 papers were considered for qualitative synthesis. Although there was heterogeneity in terms of methodological aspects, patient differences, and techniques used, the studies found that several AI approaches are promising in terms of specificity, sensitivity, and diagnostic accuracy in the detection and diagnosis of malignant tumors. When compared to other machine learning algorithms, the Super Vector Machine method performed better in cancer detection and diagnosis. Computer-assisted detection (CAD) has shown promising in terms of aiding cancer detection, when compared to the traditional method of diagnosis.

**Funding:** The funders had no role in study design, data collection and analysis, decision to publish, or preparation of the manuscript.

**Competing interests:** The authors have declared that no competing interests exist.

## Conclusions

The detection and diagnosis of malignant tumors with the help of AI seems to be feasible and accurate with the use of different technologies, such as CAD systems, deep and machine learning algorithms and radiomic analysis when compared with the traditional model, although these technologies are not capable of to replace the professional radiologist in the analysis of medical images. Although there are limitations regarding the generalization for all types of cancer, these AI tools might aid professionals, serving as an auxiliary and teaching tool, especially for less trained professionals. Therefore, further longitudinal studies with a longer follow-up duration are required for a better understanding of the clinical application of these artificial intelligence systems.

## Trial registration

**Systematic review registration.** Prospero registration number: CRD42022307403.

## Introduction

Since early diagnosis of cancer is associated with better treatment outcomes for the patient, there is substantial interest in using artificial intelligence (AI) technology in cancer screening and detection through image recognition, in the hope of reducing diagnosis times and increasing diagnostic accuracy [1]. AI has made significant advances in fields including medicine, biomedicine, and cancer research. To forecast cancer behavior and prognosis, AI employs mathematical approaches that aid in decision-making or action based on logical and autonomous thinking and effective adaptability [2–4].

AI has the potential to dramatically affect nearly all aspects of oncology–from enhancing diagnosis to personalizing treatment and discovering novel anticancer drugs. Thus, it is important to review the recent enormous progress in the application of AI and its potential in daily clinical practice, and also to highlight limitations and pitfalls for such purpose [1,2]. Several studies have attested to the potential of AI-based techniques to predict diagnosis, prognosis and response to treatment in some malignant tumors, including colorectal, breast, skin, and lung cancer [5–8].

Machine learning (ML), a branch of AI, has been shown to minimize intercurrences in dysplasia and cancer categorization, assuring uniformity and validity, and influencing treatment decisions [9]. Progress in Deep Learning (DL) approaches has shown gains in image-based diagnosis and illness detection in the study of cancer and oncology [10,11]. DL configurations are non-linear layered artificial neural networks that are hierarchically coupled. A range of DL architectures based on input data types have been developed during the last few years. Simultaneously, the model's performance was evaluated, and it was discovered that the use of DL in cancer prediction is superior than the standard procedures employed in ML [12].

In this context, these systems offer a lot of potential to support and enhance diagnostic methods, such as overcoming the limitations of human memory and attention, improving the effectiveness of computations and interpreting data, and preventing biases and prejudices from influencing judgments. However, radiologists find it challenging to assimilate and evaluate a significant volume of data to perform diagnosis and therapy because of the enormous volume and complexity of the picture data. The diagnosis takes longer, there is a higher risk of mistakes, and radiologists are prone to become fatigued. Automation in the field of radiological imaging can help to solve a number of issues, including a) improving the accuracy and

precision of picture analysis [13]; b) reducing interobserver variability [14]; and c) increasing the speed of image analysis and reports [15,16]. Thus, medical analysis demands the evolution of automated decision-making systems, with the aid of the use of computational intelligence for fast, accurate and efficient diagnosis [17], prognosis and treatment of diseases, such as brain tumors [18].

AI models, such as artificial neural networks (ANNs), have been popular in diagnostic and predictive decision-making procedures when clinical situations are complicated, such as liver cancer [19], malignant melanoma and breast cancer [20,21], and colon cancer [22]. Image processing, pattern recognition, artificial intelligence, and medical pictures are all combined in Computer-Aided Diagnosis (CADs) systems. Several computer-based solutions, such as Computer Aided Diagnosis (CADx) or Computer-Aided Detection (CADe), have been suggested to aid the radiologist in the process of interpreting computed tomography (CT) scans. CADe systems may detect and label suspicious regions as lesions in an image, while CADx systems not only highlight suspicious areas, but also point out the nature of the detected lesion as malignant or benign [23,24]. Therefore, CAD systems might potentially decrease the workload of radiologists, leading to fast and accurate diagnoses.

The terms computer-aided detection (CADe) and computer-aided diagnosis (CADx) are frequently used to describe CAD in the literature. By calling radiologists' attention to questionable areas in an image, CADe schemes aim to eliminate observational oversight. On the other hand, CADx strategies aim to classify a worrisome area and characterize it. CAD schemes and ML-based prediction models for medical images, such as breast imaging, for example, have limited therapeutic relevance despite significant research efforts and the availability of marketed CAD solutions [25]. Radiomics, on the other hand, is a discipline that has emerged as a result of the recent quick breakthroughs in bioinformatics and the introduction of high-performance computers. Radiomics includes calculating numerical image-based features that can be mined and applied to forecast clinical outcomes [26]. To measure and define the size, shape, density, heterogeneity, and texture of the targeted tumors in medical imaging, radiomic techniques are utilized to extract a large number of features from a series of medical images [27]. Segmenting the tumor region and extracting features from there is one way to guarantee that the derived features have some clinical value. As a result, manual or partially automated tumor segmentation is used in several radiomics-based systems. New methods for creating CAD schemes are also being investigated and described in the literature due to the increasing enthusiasm for deep learning-based artificial intelligence (AI) technology [28]. Numerous research have contrasted CAD schemes employing deep learning techniques and traditional radiomics to examine their benefits and drawbacks [29,30].

Since deep learning models can directly extract characteristics from medical images, DL-based CAD schemes are appealing [31]. However, despite the difficulty in achieving high scientific rigor when creating AI-based deep learning models [32], using AI technology to create CAD schemes has emerged as the research standard. Aside from cancer detection and diagnosis, new AI-based models are being broadened to incorporate extensive clinical applications such short-term cancer risk and prognosis prediction and clinical outcome.

Currently, despite systematic reviews on the subject, there is still no overview in the literature that brings together the knowledge of published systematic reviews regarding the use of artificial intelligence in cancer detection in a single publication.

Considering the current potentialities of the aforementioned AI-driven systems for the oncologic field, the capability of these systems to detect malignant tumors based on different imaging modalities should be investigated. Therefore, this overview article aims to answer the following question: When compared to standard imaging diagnosis, how accurate are artificial intelligence applications for cancer detection in adult patients?

## Materials and methods

### Protocol registration

The protocol of this study was registered on the International Prospective Register of Systematic Reviews—PROSPERO (www.crd.york.ac.uk/PROSPERO/) under number CRD42022307403. This overview was conducted according to the Preferred Reporting Items for Systematic Reviews and Meta-analyses, following the PRISMA checklist (http://www.prisma-statement.org/) and was developed according to the JBI Manual for Evidence Synthesis (https://synthesismanual.jbi.global) and the Cochrane Handbook for Systematic Reviews (www.training.cochrane.org/handbook).

The definition of systematic reviews considered was that established by the Cochrane Collaboration. A study was considered a systematic review when reporting or including:

i. research question;

ii. sources that were searched, with a reproducible search strategy (naming of databases, naming of search platforms/engines, search date and complete search strategy);

iii. inclusion and exclusion criteria;

iv. selection (screening) methods;

v. critically appraises and reports the quality/risk of bias of the included studies;

vi. information about data analysis and synthesis that allows the reproducibility of the results; [33,34]

### Search strategy

On January 21th, 2022, a broad search of articles without language or time limits was performed in the following databases: PubMed, Cochrane Central Register of Controlled Studies (Cochrane), SciVerse Scopus (Scopus), Web of Science, Latin American and Caribbean Health Sciences (LILACS), Excerpta Medical Database (Embase), Scientific Electronic Library Online (Scielo), Business Source Complete (EBSCOhost) and grey literature through Proquest, Google Scholar and JSTOR. The following Medical Subject Headings (MeSH) terms "Cancer Early Diagnosis," "Artificial Intelligence," "remote technology," "neoplasm" and synonyms were used to develop the search strategy and acquire the main strategy in PubMed. When words with different spelling appeared, synonyms that were in the MeSH terms were used. This strategy was adapted for the other databases. The search strategy used is in S1 Table. Manual searches of reference lists of relevant articles were also performed.

Immediately after literature search, the references were exported to reference manager online Rayyan QCRI (https://rayyan.qcri.org/welcome) and duplicated references were removed.

### Inclusion and exclusion criteria

PIRDs (Participants, Index test, Reference Test, Diagnosis of Interest and Studies) acronym was used to define inclusion and exclusion criteria. As inclusion criteria, diagnostic models and/or detection tool of any type of cancer in adult patients (P) in systematic reviews using AI (I) compared to the traditional model of diagnostic radiographic imaging (R) were evaluated. For the diagnoses of interest (D), the following accuracy metrics for detecting and diagnosing cancer were considered: sensitivity, specificity, Receiver Operating Characteristic (ROC) curve, and Area Under the Curve (AUC).

Exclusion criteria comprised: 1—Studies evaluating diagnosis of areas other than medicine and dentistry (Physiotherapist, Nutritionist, Nursing, Caregivers etc.); 2 –Patients with a confirmed diagnosis of cancer; 3—Systematic Reviews on AI, ML, DL and CNN not evaluating the diagnostic accuracy of the systems; 4—Systematic Reviews with AI use for other diseases diagnosis (Diabetes, Hypertension, etc); 5—Systematic reviews in which AI was not compared to a reference test; 6—Systematic reviews evaluating other technologies for detection or cancer diagnosis (spectrometry, biomarkers, autofluorescence, Multispectral widefield optical imaging, optical instruments, robotic equipment etc.); 7—literature reviews, integrative reviews, narrative reviews, overviews; 8—Editorials/Letters; 9—Conferences, Summaries, abstracts and posters; 10 —In vitro studies; 11—Studies of animal models; 12—Book chapters; 13—Pipelines, guidelines and research protocols; 14—Review papers that, despite self-styled systematic reviews, do not fulfill the criteria for the definition of Systematic Reviews; 15—Primary studies of any type.

## Data extraction

The studies selection was performed in two phases. On phase 1, two independent reviewers (HECS and GNMS) evaluated titles and abstracts of all records, according to the eligibility criteria. On phase 2, both reviewers (HECS and GNMS) independently read the full texts according to the inclusion and exclusion criteria. In case of disagreements, both reviewers discussed and, if consensus was not reached, a third reviewer (AFL) was consulted to reach a final decision. At phase 2, the articles were excluded if they did not fulfill the key characteristics of systematic reviews according to the following criteria [33,34]:

1. Those carried out by a single reviewer

2. Those who do not propose a specific research question (e.g., using PICOS or another appropriate acronym);

3. Those who do not determine pre-specified eligibility criteria;

4. Those who do not use a pre-specified search strategy;

5. Those who do not apply the search strategy to at least two databases

6. Those that do not provide a clear description of the study selection process (methods used to include and exclude research at each level);

7. Those who do not use any method (qualitative/narrative or quantitative using instruments) to assess the methodological quality of included studies.

## Study selection

Data extraction was also performed by two independent reviewers (HECS and GNMS) and crosschecked. Extracted data comprised: Author, year, country; Design of included studies; N of included Studies/ N of select studies; Type of cancer; Index test; Reference test; True positives / N of images; True Negatives /N of images; Sensitivity and Specificity/ odds ratio Mean±SD, *p-value*; Diagnostic accuracy; and main conclusions of each paper. When necessary, request for additional information, via email, was made to the authors of the selected articles. Three authors did not provide consolidated data in the form of quantitative analysis. Despite contact via email and social networks, there were no responses from any of the three authors [35–37].

## Assessing the methodological quality of included studies

The Critical Appraisal checklist for Systematic Reviews (Joanna Briggs Institute, 2014) was used to assess the methodological quality of the studies independently by two reviewers

(HECS and GNMS) [38]. It should be noted that critical appraisal/risk of bias tools classically indicated for systematic reviews, such as AMSTAR 2 and ROBIS, were designed for systematic reviews of intervention, while the articles included were systematic reviews of diagnostic accuracy. We opted for performing the methodological assessment, not the risk of bias in the selected studies.

Studies were characterized according to the scoring decisions agreed by reviewers previously. Systematic Reviews were considered of "low" methodological quality when only 1 to 4 tool items received "yes" answers; "moderate" quality with 5 to 8 "yes" answers; and "high" quality with 9 to 11 "yes" answers.

### Considered outcomes

The indexes and reference tests were compared concerning to cancer detection and diagnosis (sensitivity, specificity, ROC, AUC). Despite previously planned on the protocol, meta-analysis of the data was unfeasible due to studies' high methodological heterogeneity.

## Results

### Description of included studies

The electronic search of five databases and grey literature retrieved 382 records. Removal of 18 duplicated studies resulted in 364 records. Titles and abstracts from these studies were read and those not fulfilling the eligibility criteria were excluded. In addition, 40 records retrieved from grey literature were considered. At the end of phase 1, 32 papers remained for full text reading (phase 2). Manual search of reference lists did not provide additional studies. Full text reading resulted in 09 eligible studies for qualitative analysis. S2 Table presents excluded articles and reasons for exclusion. A flowchart of the complete process inclusion is shown in Fig 1.

Included studies were conducted in EUA [28], Netherlands [36], Italy [40], Sweden [35], China [41,42], Indonesia [43], United Kingdom [44] and Denmark [37]. All included studies were published in English. One SR included descriptive studies [39], three RS included diagnostic accuracy studies [40,43,44], four SR included prospective or retrospective studies [35,36,41,42] and one SR included clinical trial studies [37]. The accuracy of AI for detecting cancer in adult patients was evaluated by sensitivity, specificity, ROC, and AUC.

Table 1 summarizes study details regarding participants, index test, reference test, outcomes (true positive, true negative, sensitivity, specificity and diagnostic accuracy) and conclusions.

### Methodological quality within studies

None of the studies fulfilled all methodological quality criteria. However, five studies [39–42,44] were considered of "high" methodological quality, three studies [35,37,43] were of "moderate" methodological quality and only one study [36] was considered of "low" methodological quality.

In two studies [36,44], the review question was not considered clearly and explicitly stated. The inclusion criteria was not appropriate for the review question in one study [36], the sources and resources used to search for studies was not adequate in one study [39], the likelihood of publication bias was not assessed in four studies [35–37,39], the recommendations for policies and/or practices supported by the reported data were unclear for a study [37], and the specific directives for new research were inconclusive for three studies [36,37,41]. In all of studies the search strategy and the criteria for appraising studies were appropriate.

More information about the methodological quality assessment of included studies can be find in Table 2 (summarized assessment).

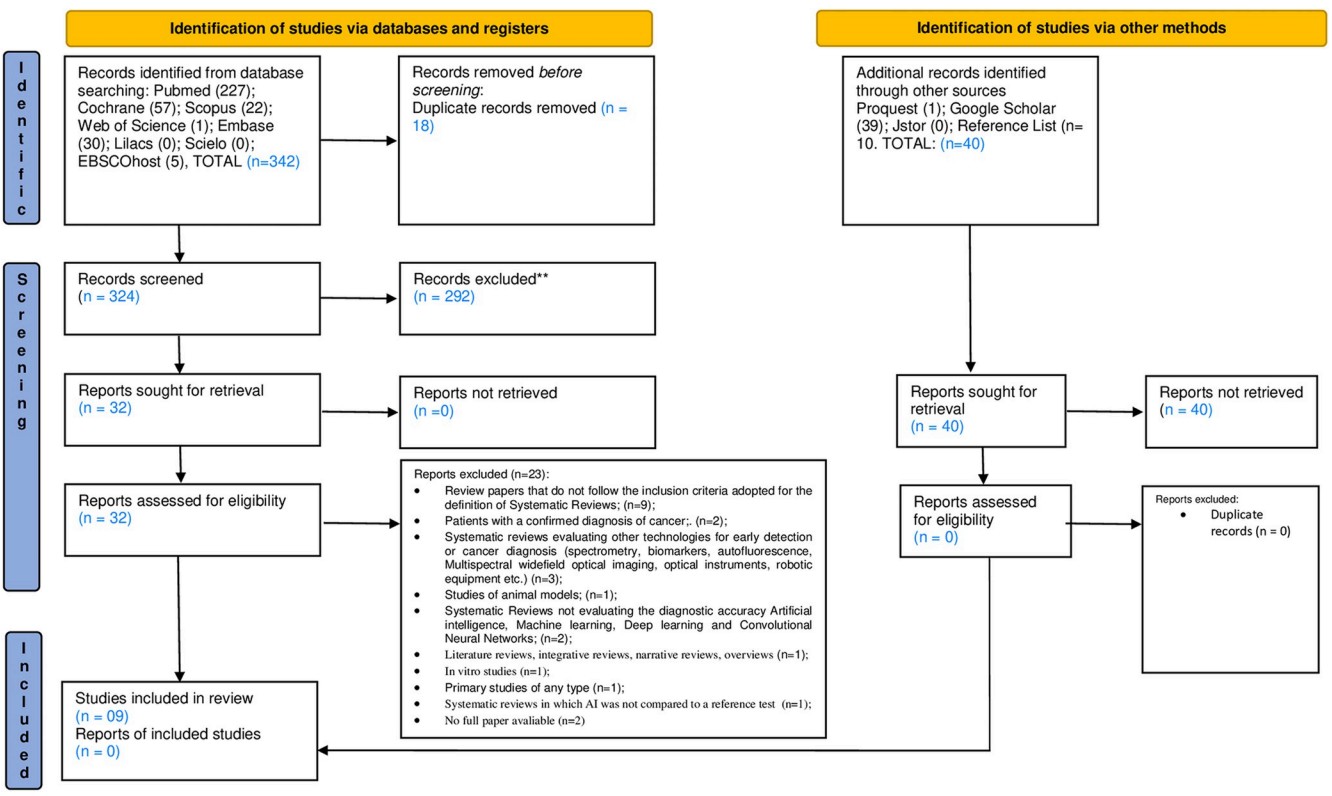

**Fig 1. Flow diagram of the literature search and selection criteria.**

## Results of individual studies

The systematic review conducted by the Department of Radiology at the University Medical Center Groningen in the Netherlands, looked at computer-assisted detection (CAD) in breast MRI and evaluated radiologists' accuracy in distinguishing benign from malignant breast lesions. Of the 587 papers assessed by the study authors, the 10 studies selected by eligibility criteria included a total of 895 patients with a total of 1264 breast lesions. Sensitivity and specificity were used to compare the performance accuracy of radiologists with and without CAD. Radiologists with experience attained a non-CAD sensitivity of 89% and a CAD sensitivity of 89%, respectively. On the other hand, the specificity was 86% without CAD and 82% specificity with CAD, respectively. Residents' sensitivity rose from 72% to 89% with CAD, while the difference was not statistically significant. In terms of specificity, the findings without CAD 79% and with CAD 78% were identical. The CAD in breast MRI has little bearing on the sensitivity and specificity of competent doctors. [39].

The reviewers from Universitas Gadjah Mada in Indonesia conducted a systematic review to establish the diagnostic accuracy of various ML algorithms for calculating breast cancer risk. There were 1,879 publications assessed in total, with 11 being included in systematic review and meta-analysis. Super Vector Machine (SVM), Artificial Neural Networks (ANN), Decision Tree (DT), Naive Bayes (NB), and K-Nearest Neighbor were identified as five types of ML algorithms used to detect breast cancer risk (KNN). The AUC of the Summary Receiver Operating Characteristic (SROC) for the SVM method was > 90%, demonstrating the greatest performance among the algorithms studied in terms of calculating the risk of breast cancer, and thus having the best precision value compared to other machine learning algorithms [43].

**Table 1. Summary of descriptive characteristics of included articles (n = 09).**

| Author, year, country and design studies | Included Studies | Type of cancer | Index test | Reference test | True positives / N of images | True Negatives /N of images | Sensitivity and Specificity/ odds ratio, Mean ±SD, *p' value* | Diagnostic accuracy (%), Mean±SD, *p' value* | Conclusions |
|---|---|---|---|---|---|---|---|---|---|
| Dorrius et al, 2011 [39], Netherlands, Descriptives studies | 10 | Breast Cancer | Computer-aided-detection (CADe) | Magnetic Resonance Imaging (MRI) | - | - | Sensitivity Radiologist no CAD, general 82% (95% CI: 72%–90%) Radiologist with CAD, general 89% (95% CI: 83%–93%) Specificity Radiologist no CAD, general 81% (95% CI: 74%–87%) Radiologist with CAD, general 81% (95% CI: 76%–85%) | - | MR images CAD has little influence on the sensitivity and specificity of the performance of radiologists experienced in breast MRI diagnosis. Breast MRI interpretation by radiologists remains essential. Radiologists with less experience seem to benefit from a CAD system when performing breast MRI evaluation. |
| Henriksen EL et al, (2018) [37], Denmark Clinical trials | 13 | Breast cancer | CAD system.; Single Reading (SR) SR vs SR + CAD; Double Reading (DR) DR vs SR þ CAD; | MM | - | - | - | - | In conclusion, all but two studies found that SR CAD improves mammography screening RRs, sensitivity, and CDR when compared to SR alone. No statistically significant variations in sensitivity or CDR were seen when compared to DR. More research is needed to assess the impact of CAD in a population-based screening program with high-volume readers. Longer follow-up studies are required for a thorough assessment of cancer rates. And studies based on digital mammography are required to assess the efficacy of CAD in the current standard of care technology. |
| Nindrea et al, 2018 [43], Indonesia, Diagnostic Accuracy studies | 11 | Breast cancer | Machine Learning Algorithms Super Vector Machine (SVM); Artificial Neural Networks (ANN); Decision Tree (DT); Naive Bayes (NB); K-Nearest Neighbor (KNN) | Mammography (MM) | SVM 40,37%/ 3532; ANN 1,30%/ 63325 DT 33,19%/738 NB 35,32%/ 1039 KNN 41%/1568 | SVM 46,40%/ 3532 ANN 97,88%/ 63325 DT 61,38%/738 NB 54,66%/ 1039 KNN 44,89%/ 1568 | Sensitivity SVM: 0.67–0.99 (95% CI: ([0.41–0.87]-[0.95–1.00]); ANN: 0.84–0.97 (95% CI: ([0.60–0.97]-[0.95–98]); DT: 0.90–0.92 (95% CI: ([0.68–0.99]-[0.88–.95]); NB: 0.76–0.91 (95% CI: ([0.68–0.83]-[0.87–.95]); KNN: 0.56–0.95 (95% CI: ([0.48–0.64]-[0.92–0.97]); Specificity SVM: 0.60–0.98 (95% CI: ([0.36–0.81]-[0.96–1.00]); ANN: 0.71–0.99 (95% CI: ([0.48–0.89]-[0.99–0.99]); DT: 0.79–0.97 (95% CI: ([0.54–0.94]-[0.9–0.98]); NB: 0.78–0.99 (95% CI: ([0.52–0.94]-[0.9–1.00]); KNN: 0.53–0.99 (95% CI: ([0.44–0.61]-[0.93–0.97]); | SVM: 99.51%; ANN: 97.3%; DT: 95.13%; NB: 95.99%; KNM: 95.27%; | Therefore, the early diagnosis of breast cancer will be more effective, and the mortality rate of breast cancer will decrease. Additionally, if the present method is designed in the form of a web-based or smartphone application, women who want to know their own risk of breast cancer will be able to access this information easily in daily life. |

*(Continued)*

**Table 1.** (Continued)

| Author, year, country and design studies | Included Studies | Type of cancer | Index test | Reference test | True positives / N of images | True Negatives /N of images | Sensitivity and Specificity/ odds ratio, Mean ±SD, *p' value* | Diagnostic accuracy (%), Mean±SD, *p' value* | Conclusions |
|---|---|---|---|---|---|---|---|---|---|
| Azavedo et al, 2012 [35], Sweden, Prospective or Retrospective studies | 4 | Breast cancer | Computer-aided-detection (CAD) | MM | - | - | - | - | The scientific evidence is insufficient to determine whether CAD + single reading by one breast radiologist would yield results that are at least equivalent to those obtained in standard practice, i.e. double reading where two breast radiologists independently read the mammographic images. |
| Eadie et al, 2012 [44], United Kingdom, Diagnostic Accuracy studies | 48 | Breast cancer, lung cancer, liver cancer, prostate cancer, bone cancer, bowel cancer, skin cancer, neck cancer. | CADe; Diagnostic CAD (CADx) | MM; Breast ultrasound (BUS); BUS + mammogram; Lung Conputered Tomography (LCT); Dermatologic; | - | - | Sensitivity (SD) CADe overall Radiologist alone: 80.41±1.46 With CAD: 84.02±1.30 CADx overall Radiologist alone: 2.79±6.12 With CAD: 90.66±4.07 Specificity (SD) CADe overall Radiologist alone: 90.10±1.97 With CAD: 87.08±2.75 CADx overall Radiologist alone: 83.00±14.46) With CAD: 88.04±15.03 | Diagnostic odds ratio (DOR) (SD) CADe overallRadiologist alone3.63±0.16With CAD:3.58±0.20CADx overallRadiologist alone3.44±0.79With CAD: 4.75±0.91 | Certain types of CAD did offer diagnostic benefit compared with radiologists diagnosing alone: significantly better ln DOR scores were seen with CADx systems used with mammography and breast ultrasound. Applications such as lung CT and dermatologic imaging do not seem to benefit overall from the addition of CAD. These findings therefore offer suggestions about how CAD can be best applied in the diagnosis of cancer using imaging. |
| Zhao et al, 2019 [42], China, Prospective or Retrospective studies | 5 | Thyroid (nodules) cancer | CADx system | US | positive likelihood ratio CADx system 4.1 (95% CI 2.5–6.9); CADx by Samsung 4.9 (95% CI 3.4–7.0); radiologists 11.1 (95% CI 5.6–21.9); | negative likelihood ratio CADx sistem 0.17 (95% CI 0.09–0.32); CADx by Samsung 0.22 (95% CI 0.12–0.38); radiologists 0.13 (95% CI 0.08–0.21); | Sensitivity CADx system 0.87 (95% CI: 0.73–0.94; $I^2$ = 93.53%); CADx by Samsung 0.82 (95% CI: 0.69–0.91; $I^2$ = 79.62%); radiologists 0.88 (95% CI: 0.80–0.93; $I^2$ = 81.66%); Specificity CADx system 0.79 (95% CI: 0.63–0.89; $I^2$ = 89.67%); CADx by Samsung 0.83 (95% CI: 0.76–0.89; $I^2$ = 27.52%); radiologists 0.92 (95% CI: 0.84–0.96; $I^2$ = 84.25%); | DOR CADx system25 (95% CI: 15–42; $I^2$ = 15.5%, $p$ = 0.315);CADx by Samsung23 (95% CI: 11–46; $I^2$ = 35.9%, p = 0 .197);radiologists86 (95% CI: 47–158; $I^2$ = 41.1%, $p$ = 0.147) | The sensitivity of the CAD system in thyroid nodules was similar to that of experienced radiologists. However, the CAD system had lower specificity and DOR than the experienced radiologist. The CAD system may play the potential role as a decision-making assistant alongside radiologists in the thyroid nodules' diagnosis. |
| Cuocolo et al, 2020 [40], Italy, Diagnostic Accuracy studies | 12 | PCa | Machine learning (ML) ANN; SVM; LDA; NB; Linear regression (LIR); Random forest (RF); Logistic regression (LOR); Convolutional neural network (CNN); Deep transfer learning (DTL); | MRI | - | - | ML in PCa identification–overall (95%CI: 0.81–0.91; $I^2$ = 92%, <0.0001); Biopsy group (95%CI: 0.79–0.91; $I^2$ = 87%, p <0.0001); Radical prostatectomy group (95%CI: 0.76–0.99; $I^2$ = 93%, p <0.0001); Deep learning (95%CI: 0.69–0.86; $I^2$ = 86%, $p$ = 0.0001); Non-deep learning (95%CI: 0.85–0.94; $I^2$ = 89%, p <0.0001); | AUC overall AUC = 0.86Biopsy groupAUC = 0.85; Rradical prostatectomy groupAUC = 0.88;Deep learningAUC = 0.78; Non-deep learningAUC = 0.90; | The findings show promising results for quantitative ML-based identification of csPCa. The results suggest that the overall accuracy of ML approached might be comparable with that reported for traditional Prostate Imaging Reporting and Data System scoring. Nevertheless, these techniques have the potential to improve csPCa detection accuracy and reproducibility in clinical practice. |

*(Continued)*

**Table 1.** (Continued)

| Author, year, country and design studies | Included Studies | Type of cancer | Index test | Reference test | True positives / N of images | True Negatives /N of images | Sensitivity and Specificity/ odds ratio, Mean ±SD, *p' value* | Diagnostic accuracy (%), Mean±SD, *p' value* | Conclusions |
|---|---|---|---|---|---|---|---|---|---|
| Tabatabaei et al, 2021 [36], USA Retrospectives studies | 18 | Glioma | DT; KNN; SVM; RF; LOR; LDA; LIR; Least Absolute Shrinkage and Selection Operator (LAS/SO); Elastic Net (EN); Gradient Descent Algorithm (GDA); Deep Neural Network (DNN) | MRI | - | - | - | - | The results appear promising for grade prediction from MR images using the radiomics techniques. However, there is no agreement about the radiomics pipeline, the number of extracted features, MR sequences, and machine learning technique. Before the clinical implementation of glioma grading by radiomics, more standardized research is needed. |
| Xing et al, 2021 [41], China, Retrospective studies | 15 | prostate cancer (PCa); Peripheral zone (PZ); Transitional zone (TZ); Central gland (CG); | CAD system.; ANN; SVM; Linear Discriminant Analysis (LDA); Radiomic Machine Learning (RML); Non—specific classifier (NSC); | MRI | SVM 42,76%/ 608; ANN 34,55%/ 301; RML 34,78%/ 738; NSC 19,41%/ 1586; PZ 51,95%/ 256; TZ 59,67%/ 186; CG 32,39%/71; | SVM 41,94%/ 608; ANN 37,54%/ 301; RML 32,60%/ 738; NPC 65,15%/ 1586; PZ 32,81%/ 256; TZ 26,34%/ 186; CG 46,47%/71; | Sensitivity: 0.47 to 1.00 0.87(95% CI: 0.76–0.94; $I^2$ = 90.3%, $p$ = 0.00) ANN: 0.66 to 0.77 SVM: 0.87 to 0.92 LDA: NR RML: 0.96 Prostate zones PZ: 0.66 to 1.00 TZ: 0.89 to 1.00 CG: 0.66 Specificity: 0.47 to 0.89 0.76(95% CI: 0.62–0.85; $I^2$ = 95.8%, $p$ = 0.00) ANN: 0.64 to 0.92 SVM: 0.47 to 0.95 LDA: NR RML: 0.51 Prostate zones PZ: 0.48 to 0.89; TZ:0.38 to 0.85; CG:0.92 | AUC 0.89 (95% CI: 0.86–0.91) | The study indicated that the use of CAD systems to interpret the results of MRI had high sensitivity and specificity in diagnosing PCa. We believe that SVM should be recommended as the best classifier for the CAD system. |

Subtitles: CADe = Computer-aided-detection; MRI = Magnetic Resonance Imaging; SVM = Super Vector Machine; ANN = Artificial Neural Networks; DT = Decision Tree; NB = Naive Bayes; KNN = K-Nearest Neighbor; MM = Mammography; CADx = Diagnostic CAD; BUS = Breast ultrasound; DOR = Diagnostic odds ratio; LCT = Lung Conputered Tomography; CDR = CAD on cancer detection rate (CDR); DR = double reading; RR = Recall Rate; Pca = Prostate cancer; PZ = Peripheral zone; TZ = Transitional zone; CG = Central gland; LDA = Linear Discriminant Analysis; RML = Radiomic Machine Learning; NSC = Non—specific classifier; ML = Machine learningA; LIR = Linear regression; RF = Random forest; LOR = Logistic regression; CNN = Convolutional neural network; DTL = Deep transfer learning; LAS/SO = Least Absolute Shrinkage and Selection Operator; EN = Elastic Net; GDA = Gradient Descent Algorithm; DNN = Deep Neural Network; SR = Single Reading; DR = Double Reading.

The systematic review carried out by researchers from the University College London, United Kingdom, searched the literature for evidence of the effectiveness of a CAD systems in cancer imaging to assess their influence in the detection and diagnosis of cancer lesions by radiologists. A total of 9,199 articles were reviewed, of which 16 papers with radiologists using CAD to detect lesions (CADe) and 32 papers with radiologists using CAD to classify or diagnose lesions (CADx) were included for analysis. CADx was observed to significantly improve diagnosis in mammography, with a diagnostic odds ratio (DOR) value of 4.99 (0.53), with an average increase of 8 and 7% between without and with CADx for sensitivity and specificity, respectively; and for the breast ultrasound DOR was 4.45 (1.40), with a mean increase of 4 and 8% for sensitivity and specificity, respectively. In cases where CADx were applied to

**Table 2. Evaluation of methodological quality of included systematic reviews (n = 9).**

| Study | Methodological quality items assessed | | | | | | | | | | | Overall quality[a] |
|---|---|---|---|---|---|---|---|---|---|---|---|---|
| | Q1 | Q2 | Q3 | Q4 | Q5 | Q6 | Q7 | Q8 | Q9 | Q10 | Q11 | |
| Dorrius (2011) [39] | N | Y | Y | N | Y | Y | Y | Y | Y | Y | Y | High |
| Nindrea (2018) [43] | N | Y | Y | Y | Y | Y | Y | U | N | Y | N | Moderate |
| Eadie (2012) [44] | N | Y | Y | Y | Y | N | Y | Y | Y | Y | Y | High |
| Zhao (2019) [42] | N | Y | Y | Y | Y | N | Y | Y | Y | Y | Y | High |
| Henriksen (2019) [37] | Y | Y | Y | Y | Y | N | Y | Y | N | U | U | Moderate |
| Azavedo (2012) [35] | N | Y | Y | Y | Y | Y | U | N | N | Y | Y | Moderate |
| Cuocolo (2020) [40] | N | Y | Y | Y | Y | Y | Y | Y | Y | Y | Y | High |
| Xing (2021) [41] | N | Y | Y | Y | Y | Y | Y | Y | Y | Y | U | High |
| Tabatabaei (2021) [36] | N | U | Y | Y | Y | U | U | U | N | Y | U | Low |

Note: JBI Critical Appraisal Tool for Systematic Reviews—Q1. Is the review question clearly and explicitly stated? Q2. Were the inclusion criteria appropriate for the review question? Q3. Was the search strategy appropriate? Q4. Were the sources and resources used to search for studies adequate? Q5. Were the criteria for appraising studies appropriate? Q6. Was critical appraisal conducted by two or more reviewers independently? Q7. Were there methods to minimize errors in data extraction? Q8. Were the methods used to combine studies appropriate? Q9. Was the likelihood of publication bias assessed? Q10. Were recommendations for policy and/or practice supported by the reported data? Q11. Were the specific directives for new research appropriate?
[a]Low quality: 1 to 5 "yes" answers; Moderate quality: 6 to 10 "yes" answers; High quality: 11 to 13 "yes" answers
Abbreviations: N, no; U, unclear; Y, yes.

pulmonary CT, DOR was 2.79 (1.45) and to dermatological images DOR was 3.41 (1.00). It was found diagnostic contradictions with a mean decrease in specificity on pulmonary CT of 7% and on dermatological images of 17%. There was no evidence of benefit from using CADe. The review showed that CADx may offer some benefit to radiologists in specific imaging applications for breast cancer diagnosis although there is no evidence that it can be used in a generalized way, suggesting its application in some types of cancer diagnosis [44].

Based on a study of the current literature, reviewers from Sichuan University in Sichuan, China, conducted a meta-analysis to determine the accuracy of CAD for thyroid nodule diagnosis. A total of 1,206 publications were screened, with 5 of them being chosen for systematic review and meta-analysis in a set of 536 patients and 723 thyroid nodules. The CAD system's sensitivity in diagnosing thyroid nodules was 0.87, which was comparable to expert radiologists' 0.88. However, the CAD system had lower specificity of 0.79 and DOR of 25 when compared to specificity of 0.92 and DOR of 86 of experienced radiologists. The CAD system has potential as an auxiliary tool in decision making, being a possible ally of radiologists in the diagnosis of thyroid nodules [42].

The accuracy and recall rates (RR) of single reading (SR) vs SR + CAD and double reading (DR) vs SR + CAD were examined in a systematic study undertaken by authors from Metropolitan University College in Copenhagen, Denmark. They looked at 1,522 papers of which 1,491 were excluded by abstract. Of the remaining 31 articles, 18 were excluded after full text reading, and therefore 13 matched the review's inclusion criteria. Except for two publications in the SR vs. SR + CAD comparison, adding CAD increased sensitivity and/or cancer detection rate (CDR). There were no significant variations in sensitivity or CDR between the DR group and the SR + CAD group. In all but one research, adding CAD to SR raised RR and lowered specificity. Only one study found a significant difference between the DR and SR+CAD groups. To assess the efficacy of CAD, more research is needed based on coordinated population-based screening programs with extended follow-up times, high-volume readers, and digital mammography [37].

Researchers from Lund University, Skne University Hospital Malmö, Sweden, conducted a systematic review to verify whether readings of mammographic images by a single breast radiologist plus CAD were at least as accurate as readings by two breast radiologists. The authors looked over 1,049 papers of which 996 were excluded. 53 full-text articles were assessed for eligibility and only four met the inclusion criteria, with a population of 271,917 women being investigated. The findings suggested that there was inadequate scientific evidence to establish whether a single mammography reading by a breast radiologist plus CAD is as accurate as the present method of double reading by two breast radiologists. Similarly, the scientific evidence in the literature was insufficient to investigate cost-effectiveness, and the study's quality was deemed low [35].

Authors from the Italian University of Naples "Federico II" conducted a systematic evaluation to assess the diagnostic accuracy of ML systems for diagnosing prostate cancer (csPCa) using magnetic resonance imaging. After the final editing, a total of 3,224 articles were evaluated, of which 3,164 were excluded. Thus, 60 full-text articles were blindly evaluated by each investigator for eligibility, with 12 articles included, with a total of 1979 imaging screenings evaluated. As in the general analysis, statistical heterogeneity was considerable in all subgroups. In the identification of csPCa, the overall AUC for ML was 0.86. The AUC for the biopsy subgroup was 0.85. The AUC for the radical prostatectomy subgroup was 0.88 and Deep learning had an AUC of 0.78. The systematic review presents promising results for the quantitative identification of csPCa based on ML, with the potential to generate improvements in the detection of csPCa in terms of accuracy and reproducibility in clinical practice [40].

The diagnosis accuracy of CAD systems based on magnetic resonance imaging for PCa was investigated in a systematic review conducted by Gansu University of Traditional Chinese Medicine in China. A total of 3107 articles were examined. Of these, 3070 were excluded and of the remaining 37 articles, 15 were included for analysis with a total of 1945 patients. The overall sensitivity of the CAD system varied from 0.47 to 1.00, with specificity ranging from 0.47 to 0.89, according to the meta-analysis. The CAD system's sensitivity was 0.87, specificity was 0.76 and AUC was 0.89. Among the CAD systems, the SVM exhibited the best AUC, with sensitivity ranging from 0.87 to 0.92 and specificity ranging from 0.47 to 0.95. In terms of prostate zones, the CAD system exhibited the highest AUC in the transitional zone, with sensitivity ranging from 0 to 1. The review points out the advantage of using CAD systems for prostate cancer detection due to its high sensitivity and specificity, and the best performance of SVM algorithm for the aforementioned detection purpose [41].

The authors of a systematic review undertaken by the University of Alabama at Birmingham (UAB), Birmingham, AL, USA, analyzed the most current studies in the classification of gliomas by radiomics based on machine learning, evaluating the clinical utility and technical flaws. At the end of the screening phase, a total of 2858 patients were analyzed, from 18 articles that were chosen from 1177 publications, with 1159 papers excluded in the selection process according to the eligibility criteria adopted. The results were promising for predicting the quality of MRI images using radiomics approaches. However, there was no consensus on the radiomics pipeline, considering that the selected articles have employed a wide range of software, large amount of extracted features, different sequences and machine learning techniques. As a result, the authors urge that more standardized research should be done before radiomic glioma categorization is used in clinical practice [36].

## Certainty of the evidence in the systematic review's included

Only two articles [35,41] used the Grading of Recommendations Assessment, Development, and Evaluation (GRADE) method to assess the evidence, which examines five factors: risk of

bias, indirectness, inconsistency, imprecision, and publication. Due to the risk of bias and inconsistency, one paper [41] discovered low quality evidence for the following outcomes: true positives (patients with prostate cancer), true negatives (patients without prostate cancer), false negatives (patients incorrectly classified as not having prostate cancer), and false positives (patients incorrectly classified as having prostate cancer).

The second systematic review [35] evaluated only one study regarding the certainty of evidence for the following outcomes: Cancer detection rate and Recall rate, and the quality of the evidence found was very low due to the risk of bias and Indirectness.

## Overlapping

Within the RS reviews, included in this overview, a total of 136 primary studies were found. Approximately 3.67% of these main studies were included in multiple SRs. Only five studies were mentioned more than once. S3 Table provides more details on the overlap and features of the primary studies.

## Discussion

To the best of the authors' knowledge, this is the first overview article that critically appraise the scientific evidence of AI use for detecting and diagnosing malignant tumors on different imaging modalities. As this is a current and relatively novel topic, nine recent published SRs were retrieved in the literature search. These SRs found high accuracy metric results for the aforementioned diagnostic purpose, demonstrating the potential of AI tools for the oncologic field. The selected studies demonstrated the use of computer-assisted detection (CAD) [35,37,39,41,42,44], machine learning algorithms [40,41,43]and radiomic analysis [36] for detection and diagnosis of malignant tumors based on radiological images.

AI-driven methods for detecting and diagnosing cancer were analyzed by accuracy metrics, such as sensitivity, specificity, AUC, and ROC. The SVM algorithm showed better performance in the detection and diagnosis of prostate cancer and breast cancer when compared to other machine learning algorithms [41,43]. In four studies, CAD systems demonstrated some benefit in helping to detect cancer [39,41,42,44]. Nevertheless, the use of this tool did not present evidence that it can be used in a generalized way, with better indication for some types of cancer, such as breast cancer [44]. In addition, two studies found promising evidence on the use of ML and radiomic analysis in prostate cancer detection and glioma classification, with potential applicability in clinical practice [36,40].

Two questions that were often addressed in the selected articles were which professional can benefit most from the use of AI systems and how these tools should be used. The CAD systems demonstrated high values of sensitivity and sensitivity for diagnosing prostate cancer and this performance may be related to the location of the tumor in the prostate, for example, central gland, peripheral zone and transition zone. It was observed that the sensitivity and specificity in the transition zone was higher than in the peripheral zone and in the central gland [41]. Some papers corroborate the findings that radiologists benefit most from the use of CAD systems in the detection of prostate cancer lesions [45–48].

However, in other study, less experienced radiologists benefited more from the use of artificial intelligence than experienced professionals [39]. Residents or radiologists with little or no experience had greater sensitivity when accompanied by a CAD system for discriminating between breast lesions on MRI. On the other hand, the performance of experienced radiologists showed a non-significant decrease in specificity from 86% (95% CI: 79–91%) without CAD to 82% (95% CI: 76–87%) with CAD. This observation is due to the fact that CAD systems are based only on the dynamics of enhancement, without considering the morphology of

the lesion, which suggests that experienced radiologists may be misled by the enhancement pattern of CAD, resulting in decreased specificity [39]. The literature agrees with the findings that less experienced radiology professionals and residents benefit most from the use of CAD systems in the detection of lesions. [49–52]. Another study demonstrated that when evaluating thyroid nodules for malignancy using ultrasound imaging, a CAD system had similar sensitivity and negative likelihood ratios compared to experienced radiologists [42].

Two studies [35,37] found no significant evidence regarding sensitivity, specificity, and diagnostic accuracy, between single-reading or double-reading mammography compared with single-reading plus CAD or double-reading plus CAD. The use of CADe to detect lesions on images added less value to radiologists than CADx, used to diagnose lesions, with a small increase in weighted mean sensitivity but a decrease in mean specificity However, CADx did not improve diagnosis in combined mammography and breast ultrasound systems. Thus, CADx can be help radiologists that are looking for breast cancer in mammograms or ultrasounds, but it cannot be assumed that its use may be generalized, with applications in other types of cancer [44].

The literature is still controversial regarding the issue of single reading with the presence of CAD and double reading. A previous study found equivalent performance of CAD systems when a single reading was compared to double reading in the detection of cancer lesions [53]. However, for detecting pulmonary nodules, the performance of a CAD system was comparable to a second opinion reading [54]. However, there are works that demonstrate that the single reading of a reader with the help of the CAD as a second reader produces a significantly higher sensitivity than the single reading and the simulated conventional double reading, being a valuable tool for the detection of pulmonary nodules and can be used as a second opinion reading [54]. As there are also works that attest that the independent double reading produces a better detection performance, the presence and probability of CAD mass markers can improve the interpretation of mammography [55,56].

On the other hand, a recent study stated that the quality and amount of the evidence on the use of AI systems in breast cancer screening is still far from what is needed for its incorporation into clinical practice. In screening programs, AI systems are not sufficiently specialized to take the position of radiologist double reading. Larger research do not confirm promising outcomes from smaller ones [57].

Support vector machines (SVM) exhibited the best AUC among the CAD system classifiers for the detection of prostate cancer (CaP) in magnetic resonance imaging, with a range of 0.47 to 1.00 and specificity of 0.47 to 0.89, with an AUC of 0.89 (0.86–0.91). The AUC curve demonstrated stronger sensitivity and specificity in the transition zone than in the peripheral zone and the core gland of the organ, according to the location of the tumor in the prostate. As a result, the sensitivity of different regions of the human body to screening methods may be explained. Other screening methods, with the exception of CAD-assisted MRI, may not detect it due to limited sensitivity [41].

In another study, SVM was compared to four additional classification algorithms: artificial neural network (ANN), decision tree (DT), naive bayes (NB), and K-Nearest Neighbor (KNN). In the breast cancer risk calculation, SVM was shown to generate the best area under the curve (AUC), with AUC > 90%. The SVM has a 97.13% accuracy rate, demonstrating its effectiveness in predicting and detecting breast cancer and having the greatest accuracy and low error rate. In this approach, the SVM algorithm can predict breast cancer risk and outperforms other algorithms in terms of accuracy. Different machine learning algorithms, on the other hand, can aid in the diagnosis of breast cancer. They serve to decrease the risk of errors caused by weariness or inexperienced professionals, and they allow medical data to be analyzed in less time and with more precision [43].

With a combined AUC of 0.86, machine learning paired with radiomics demonstrated excellent results in the characterization of prostate cancer (csPCa). Deep learning analyses, on the other hand, were less accurate than artisanal radiomics and non-deep ML techniques, with AUCs of 0.78 and 0.90, respectively. While deep learning excels with big datasets with hundreds or even millions of examples, this is rarely the case in medical image analytics. In this case, the datasets are often made up of hundreds of patients at most, and the artisan technique outperforms deep learning in this scenario. As deep learning is also computationally more expensive and less understandable, it should be used with caution in medical image analysis and only when it significantly outperforms alternative approaches [40].

The radiomic study of gliomas using radiomic feature extraction in conjunction with various forms of machine learning has yielded encouraging findings with high sensitivity, specificity, accuracy, and AUC. Radiomics systems that used an external dataset had AUCs of 94% and 72%, respectively, indicating a more realistic performance [35]. The ability to translate DL models into real-world applications, in order to improve acceptance and the performance of DL clinically applied by physicians through the generalization of its applications, the interpretability of its algorithms, access to data, and medical ethics, is one of the challenges for the future of AI use in the medical field, particularly oncology, regarding the diagnosis and detection of cancer. The process of application generalization involves building a multimodal model using information other than the evaluated image itself, such as sample size, age, sex, ethnicity, incomplete data collection and a lack of a standard clinical protocol, clinical manifestations, laboratory tests, image data, and epidemiological histories. Due to the complexity of neural networks and the use of these unrepresentative datasets, overfit models that do not generalize to other populations and biased algorithms are produced [58].

The capacity of algorithms to do activities that call for intelligence is referred to as artificial intelligence. Machine learning is a subset of AI, and it refers to algorithms that learn from data in order to perform better. There are two ways that data given into an ML program may be represented: as features or as raw data. Lesion length is an example of a feature, which is a variable in data that may be measured. Digital mammography (DM), ultrasound (US), and magnetic resonance imaging (MRI) scans are examples of raw data in cancer imaging [59].

Learning features poses a challenge for these algorithms even though they often outperform handcrafted features in terms of performance. The subset of ML methods known as DL can be used to overcome this issue. The ability to recognize complicated patterns is the strength of machine learning and deep learning based approaches. Through feature engineering or feature learning, more detailed picture attributes, such as texture, form, border, location, etc., may be acquired. Higher accuracy can be achieved by segmentation based on detailed picture properties. By categorizing picture blocks of a particular size using a sliding window, typical machine learning based algorithms (such as RBFNN, SVM, etc.) get the whole segmentation image. This leads to unnecessary computation, misclassification, and jagged segmentation borders. On the other hand, deep learning-based approaches (such 3D U-Net CNN) outperform conventional machine learning-based methods in terms of performance and segmentation. Deep learning-based methods have greater discriminating abilities in pixel categorization because they can learn more useful picture attributes. However, many machine learning-based approaches require a large amount of labelled training data [59–61].

Features are represented in terms of other, more basic features in DL. Since DL algorithms are made up of many (deep) layers of linked neurons, they are sometimes referred to as deep neural networks (DNNs). CNNs are a specific kind of DNN. CNNs are frequently employed in cancer image analysis since they were created particularly to detect important characteristics in pictures [62,63]. Different criteria are employed for various activities in order to compare the performance of DL networks with human standards. The metrics used in categorization

are founded on receiver operating characteristic analysis. AUC, accuracy, sensitivity, and specificity all have a significant impact in this situation. Thus, accuracy represents the proportion of correctly classified samples, sensitivity represents the likelihood that the model or radiologist will output a positive (and thus malignant) result if the sample is malignant, specificity represents the probability that the model or radiologist will output a negative (and thus benign) result if the sample is benign, and AUC represents the average sensitivity for all possible specificity values [60,61].

Oncologists find it challenging to comprehend how DL models assess data and make judgments since the sheer number of parameters involved make it challenging for professionals to interpret algorithms. Data access and quality are frequently negatively impacted by a deficient data sharing network, as well as competition between different institutions. Building an open data-sharing platform with the participation of numerous institutes is the first step in overcoming these challenges. Governments and businesses must create a formal structure in the future to enable secure data sharing. Examples include privacy-preserving distributed DL (DDL), which offers a way to protect privacy and enables several participants to train jointly using a deep model without explicitly sharing local datasets. Additionally, the Cancer Imaging Archive, which compiles clinical images from many hospitals and institutes, is another excellent illustration of data sharing and can support radiomic studies [58,64,65].

Due to the need to preserve patient information, which can lead to overfitting, it is challenging to get the data in sufficient quantities to have credibility in training and validation in DL. Companies handling this data must adhere to current data protection and privacy laws in both their home countries and the countries of residence of the data subjects. Before exploiting delicate data, such as genetic data, informed agreement from patients must be sought. Patients must be informed about the potential uses of their data, and it must be made sure that everyone would benefit from them. Furthermore, thorough monitoring and validation procedures must be implemented in order to evaluate AI performance across various applications [58,64].

Before DL techniques are used in therapeutic settings, there are significant ethical issues that need to be resolved. The level of supervision needed for doctors must first be decided. Second, the party accountable for DL tools' inaccurate judgments must be identified. Before AI is implemented in real-world settings, it is also necessary to outline legal obligations in the event of a malfunction. In addition, the majority of high-end AI software works in a "black box" testing environment, meaning that users are unaware of the software's fundamental workings. The tester just knows the input/output; the reasoning behind coming to a particular conclusion is still a mystery. Clinicians frequently confront moral conundrums when making predictions without a thorough grasp of the processes underlying them, hence it is imperative to offer greater transparency in AI models by creating techniques that let users examine the details of the input data that affected the result. closer to the truth [58–65].

The main databases used in training ML and DL technologies vary according to the type of cancer. The most used ones are: Breast Cancer dataset (WBCD); Wisconsin Diagnostic Breast Cancer (WDBC); Wisconsin Prognostic Breast Cancer (WPBC) [66]; Digital Database for Screening Mammography (DDSM) [67]; The Mammographic Image Analysis Society (MIAS) [68]; Breast Cancer Digital Repository (BCDR) [69]; The Cancer Imaging Archive (TCIA) Public Access [70] and Lung Image Database Consortium–the LIDC [71]. Breast cancer databases and other databases have been reported up to date for studying cancer, but the information contained in these databases frequently presents some unfavorable issues: a) some are lacking in terms of available features (image-based descriptors, clinical data, etc.); b) others have a limited number of annotated patient cases; c) and/or the database is private and cannot be used as a reference, which makes it difficult to explore and compare performance [69]; the

lack of larger datasets with manual malignancy annotations and diagnostic cancer labels constitutes the main limitation [72].

Other limitations of the databases that can be listed are: the availability of patient-based pathologic diagnoses for only a subset of cases, the inability to perform reader studies because the files do not maintain radiologists identities or a consistent ordering of radiologists marks, the interpretation of CT scans using only transaxial images, the somewhat artificial nature of the lesion categories relative to clinical practice, the interpretation of every case is not performed by the same radiologists, and the design of the manual QA process that focus mostly on the visual identification of objective lesion annotation errors and did not analyzes inconsistencies in the subjectives lesions characteristic ratings, although the benefit of this quality assurance process to the integrity of the Database should not be understated [73].

## The critical analysis of meta-analyses that presented complete data

Of the studies selected in this overview, only three studies presented meta-analyses regarding the sensitivity, specificity and diagnostic accuracy of the use of medical radiological images in the detection of cancer lesions, based on artificial intelligence tools [39,42,44].

Critical analysis of the meta-analysis for diagnosing thyroid nodules based on ultrasound imaging through CAD [42] showed that the CAD system had similar sensitivity and negative likelihood ratio compared to experienced radiologists. However, specificity, positive likelihood ratio and DOR were relatively low. These results indicated that there was a clear gap between the CAD system and the radiologist experienced in making the diagnosis of thyroid nodules. Furthermore, successful nodule segmentations were important and influenced the nodule recognition accuracy. Nodule malsegmentation occurred more frequently with benign nodules (n = 11, 18.6%) than with malignant nodules (n = 2, 4.7%) and the difference was statistically significant (P = 0.04). Among nodules with poor segmentation, 54.6% of benign nodules (6/11) were also diagnosed as malignant, while all malignant nodules were diagnosed as malignant. As a result, it is clear that a CAD system's subpar segmentation can raise the false positive rate while having no impact on the false negative rate. The CAD system's sensitivity to thyroid nodules was comparable to that of skilled radiologists. However, compared to an expert radiologist, the CAD system showed worse specificity and DOR. [42].

Meta-analysis for the evaluation of breast lesions with MRI showed that the combined sensitivity and specificity of the experienced radiologist remain comparable with the implementation of CAD. Less experienced residents or radiologists seemed to achieve greater sensitivity with CAD implantation, although not statistically significant. Residents or radiologists with little or no experience obtained greater sensitivity when accompanied by a CAD system for discrimination of breast lesions on MRI. The change in sensitivity after using the CAD was not statistically significant. However, a considerable increase could be observed (72% sensitivity; 95% CI: 62–81% to 89%; 95% CI: 80–94%). This rise could be attributable to the fact that CAD alerts radiologist trainees or less skilled radiologists to more enhanced lesions, which may be helpful when assessing breast lesions with MRI [39].

The performance of experienced radiologists showed a non-significant decrease in specificity from 86% (95% CI: 79–91%) without CAD to 82% (95% CI: 76–87%) with CAD. A clarification for this observation may be that CAD systems are based only on the dynamics of enhancement, without taking into account the morphology of the lesion. As a consequence, the use of CAD could lead to a greater number of enhanced lesions, part of which could be classified as benign based on morphology [39].

In another study using mammograms and breast ultrasound imaging in the evaluation of CAD systems, certain types of CAD offered diagnostic benefits compared to radiologists

diagnosing alone: significantly better ln DOR scores were seen with CADx systems used with mammography and breast ultrasound. This fact can be observed, since the use of CADx tends to increase sensitivity and specificity in mammography (mean increase of 8 and 7% between without and with CADx for sensitivity and specificity, respectively) and breast ultrasound (mean increase of 4 and 8% for sensitivity and specificity, respectively), but adversely affects specificity in lung CT (mean reduction 7%), combined breast ultrasound and mammography systems (mean reduction 12%) and dermatologic imaging (mean reduction 17%). According to evidence, using CADe systems results in a tiny net overall drop in ln DOR as well as a similar-sized gain in sensitivity and loss in specificity [44].

It is also noticed that the use of CADx improved the diagnosis. However, the overlapping of the 95% confidence interval (CI) curves suggests that the difference is not significant. The AUC is 0.88 (SD: 0.03) for radiologists alone and 0.92 (SD: 0.03) for the same radiologists using CADx and 0.85 (SD: 0.19) for radiologists alone in studies of detection and 0.84 (SD: 0.19) for those radiologists using CADe [44].

The examined meta-analyses did, however, have several drawbacks. First, all displayed significant variation among trials in terms of sensitivity and specificity. This variability is probably due to both the fundamental variations in the patients who were included in the studies' methodologies. Second, the included studies' sample sizes were somewhat modest [39,42,44]. When conducting the meta-analyses, the authors took into account the possibility of selection [39,42,44], measurement [42], and publication [42,44] bias.

## The role of explainable artificial intelligence in DL and ML models

Recent advances in ML have sparked a new wave of applications for AI that provide significant advantages to a variety of fields. Many of these algorithms, however, are unable to articulate to human users why they made certain decisions and took certain actions. Explanations are necessary for users to comprehend, have faith in, and manage these new artificially intelligent partners in the crucial knowledge domains of defense, medical, finance, and law for exemplo [74–76].

New ML methods including SVMs, random forests, probabilistic graphical models, reinforcement learning (RL), and DL neural networks are significantly responsible for the current strong performance of AI. These models exhibit good performance, but they are difficult to understand. In many cases, the most performing methods (such as decision trees) are the least explainable, and the most explainable methods (such as DL) are the least accurate. Explanations might be complete or incomplete. Full explanations are provided by fully interpretable models in a transparent manner. Partially interpretable models shed light on key aspects of their thought process. Contrary to black box or unconstrained models, interpretable models adhere to "interpretability restrictions" that are established according to the domain [77].

Although there may be many different types of users, frequently at various times in the development and use of the system, the Explainable Artificial Intelligence (XAI) assumes that an explanation is provided to an end user who depends on the decisions, recommendations, or actions produced by an AI system. For instance, an intelligence analyst, a judge, an operator, developers or test operators, or policy makers. Each user group could have a particular explanation style that they find to be the most successful in conveying information [77,78].

The effectiveness of an explanation has been evaluated and measured in a number of ways, but there is presently no accepted method of determining if a XAI system is more user-intelligible than a non-XAI system. Task performance may be a more objective indicator of an explanation's efficacy than other of these indicators, such as user satisfaction. It remains an

outstanding research question how to accurately and consistently measure the impact of explanations [79,80].

Before explainability can be achieved in DL models, there are still several open problems and obstacles at the intersection of ML and explanation. First, there is a lack of consensus over the terminology and many definitions used in relation to XAI. Since XAI is still a relatively new field, there isn't yet a set of accepted terms in use [81].

Second, there is a trade-off between accuracy and interpretability [82], i.e., between the thoroughness of this description and the simplicity of the information provided by the system regarding its internal functioning. This is one of the reasons why developing objective measurements for what makes a good explanation is difficult with XAI.

Utilizing findings from experiments in human psychology, sociology, or cognitive sciences to develop objectively compelling explanations is one way to lessen this subjectivity. This would allow programmers to design software for their target audience rather than for themselves, with the evaluation of these models being more concerned with people than with technology [83,84]. A promising approach to solving this problem is to combine the connectionist and symbolic paradigms [85–89]. Connectionist approaches are more exact but opaque on the one hand. Symbolic approaches, on the other hand, are more easily understood while being generally seen as less effective. Additionally, it has been demonstrated that the introduction of counterfactual explanations might aid the user in comprehending a model's conclusion [90–92].

Third, XAI approaches for DL must address the issue of delivering explanations that are understandable to society, decision-makers, and the legal system as a whole. In order to address ambiguities and establish the social right to the (not yet existing) right to explanation under the General Data Protection Regulation of all countries in general, it will be especially important to communicate explanations that require non-technical competence [93].

It is obvious that incorporating this work into explainable AI is not an easy process. These models will need to be improved and expanded from a social science perspective in order to produce good explanatory agents, necessitating strong collaboration between explainable AI researchers and those in philosophy, psychology, cognitive science, and human-computer interaction [83].

## The use of uncertainty quantification approaches in medical imaging

In addition to using uncertainty quantification (UQ) approaches for medical image analysis, XAI is also used in decision-making in DL methods. Tools have been created to quantify the predicted uncertainty of a specific DL model (Abdar et al., 2021a). The implementation of a deep learning algorithm for uncertainty quantification in oncology can aid in improving performance while analyzing medical images. As a result, for exemplo, the outcomes of prostate cancer segmentation from ultrasound pictures are enhanced by the addition of uncertainty quantification [94].

Numerous advantages result from improving the application of the uncertainty quantification metric. In a medical setting, it becomes essential to identify questionable samples that require human evaluation in order to avoid silent errors that could result in incorrect diagnosis or treatments. Second, UQ makes it possible to spot the model's flaws, such as uncertain forecasts, which may point to a deficient training set. Inconsistencies in the incoming data might also be shown by a high level of UQ, which is crucial for quality control (QC). Overall, UQ strengthens user confidence in the algorithm and makes it easier for the algorithm and user to communicate. Additionally, UQ is supported by solid theoretical underpinnings and has developed as a clinically expected characteristic of an applied AI system [95]. In this situation, the model's predicted performance alone is insufficient to achieve a high level of acceptability.

In order to encourage human-machine collaboration and eliminate the black-box effect, UQ is essential.

In this context, the collaboration between researchers in medicine and artificial intelligence is one future study area that might be taken into consideration. As a result, the suggested machine learning and deep learning methods can do a better job of forecasting various diseases and cancers. This can be very beneficial for resolving uncertainties [94].

The collecting of medical data to the greatest extent possible is one of the gaps for enhancing the uncertainty metric in choices. The accuracy of the findings generated from medical picture segmentation depends heavily on the use of ground truth data. The sending of inaccurately projected facts to experts also plays a significant part in coping with uncertainty. Therefore, in the field of medical picture segmentation, there is a need for strong cooperation between researchers in medicine and computer science [94].

Big medical data collection may be a significant future direction. More data can significantly enhance the performance of several deep learning techniques. Transfer learning approaches, however, can be a good alternative if huge datasets are not available for training [94]. The majority of the UQ approaches that have been put into practice (81.15%) are based on a sampling protocol and try to produce several predictions for the same query input. The potential of deterministic UQ approaches that only require one step to compute uncertainty should be thoroughly investigated [96].

And finally, while being critical in real-world medical circumstances, the detection of Out-of-distribution (OOD) predictions using uncertainty is a subject of relatively few investigations. In an automated medical picture pipeline, input samples may show a variety of anomalies and artifacts that could interfere with the NN's performance and lead to severely inaccurate predictions. This inspires the creation of feature-based techniques designed specifically for OOD detection. Noting that OOD detection is a very active research area that is not exclusive to the UQ sector, it should be noted that OOD detection is currently not often used for medical picture analysis [96,97].

## Limitations of the included systematic reviews and the overview

Regarding the limitations presented in the systematic reviews included in this overview, it was observed: short follow-up time, which leads to an overestimated sensitivity [35] or a loss in the calculation of diagnostic accuracy measures [37]; relatively low number of studies [40]; high heterogeneity can be partly explained by the diversity of methodological aspects, difference between patients, or diversity of techniques used [35,40,42]; presence of selection bias by choice of articles reporting sensitivity and specificity results [44], by use of retrospective studies, vaguely reported sample of patients [35], by use of studies with relatively small samples [41,42]; possible presence of publication bias due to lack of studies with unfavorable data [44]; use of digitized analog radiographs to the detriment of digital images [35,37]; behavior of radiologists in terms of training, conducting clinical tests and surveillance in the analysis [35,44]; relatively small and old technology dataset number[35,39]; presence of measurement bias due to the large difference between the groups studied and the small number of outcomes observed in the included studies.

This overview presented as limitations: there are still few studies that use artificial intelligence, in its various approaches, in the detection of cancer, being limited to some more favorable types of cancer, such as breast cancer, prostate cancer and thyroid cancer. There is considerable heterogeneity in the methodologies of the studies, which makes it difficult to standardize the artificial intelligence technologies used. Finally, the limitation of the type and quality of images makes it difficult or impossible to use artificial intelligence in the detection of certain types of cancer, such as in the case of skin or lung cancer.

## Mains research gaps and future ML/DL research directions

The fact that deep learning algorithms demand a lot of data, sophisticated imaging technology, top-tier statisticians, and research funding to produce is one of their key gaps. First of all, because of the research's existing variances in sample size, research design, data source, and imaging collecting criteria, it is challenging to quantify, integrate, and extrapolate the findings in a way that was applicable to all situations. Additionally, researchs might exhibit a significant degree of publication bias, especially when they lack external validity [98].

Furthermore, Most AI models also disregard social and cultural risk variables, and the majority of those that have been developed were built using data from the entire population. To increase the accuracy of current models' predictions and modify these tools to the unique characteristics of the population being examined, combining critical risk factors, including imaging, pathology, demographics, clinical data, smoking status, tumor histology and new and ancient technology is advised [98,99]. Researchers can create predictive models by combining several features [100,101]. The concept of multi-omics [102,103] or "Medomics" [104] is introduced as a result. Therefore, it will be worthwhile to continue to pursue the merger of various domain expertise and multidisciplinary integration.

The need for additional large-scale multicenter prospective researchs is highlighted by the fact that this type of research necessitates big datasets. Future research should concentrate on creating deep learning models from decentralized, nonparametric data [105,106]. When compared to conventional models, these methods directly process the raw data, which reduces variability while enhancing model performance [98].

However, Large datasets on the order of (tens of) thousands of patients from various medical centers are now available for research using digital mammography (DM) and digital breast tomosynthesis (DBT). Rarely does MRI research involve more than 500 individuals, and it often comes from a single center. This certainly benefits AI performance in DM and DBT research, as larger datasets and data from various sources typically result in DL models that perform better and have better generalization. Although there are currently a number of sizable retrospective and multi-reader studies for the evaluation of DL CAD systems for DM and DBT, there are less of them for ultrasound (US) and none that that are known for MRI. Thus, DL research in US and MRI needs to invest in generating larger and more diverse datasets to move from proof-of-concept models to systems ready for large multi-case studies with multiple readers, as is now the case with DM/ DBT. However, this does not mean that all DM/DBT models are sufficiently tested for implementation in clinical practice [59,106].

In this way, sharing data between medicals center is a simple way to prevent small datasets from becoming obsolete and large ones from expanding quickly. Regulated data exchange is unfortunately a major barrier for researchers [105]. Swarm learning, where all participants contribute to both case collection and algorithm development [107,108], or even federated learning, where the data stays local but the algorithm travels [105,109,110], are positioned to solve this issue. Such methods haven't, however, been widely used up until now. The challenge of validating the precise results of DL research in cancer pictures, which is typically not achievable since the (training) data are not can be shared, is solved by the construction of checklists, showing the basic requirements for the transparent reporting AI clinical investigations. Future studies on AI will be able to be more thorough and consistent thanks to these lists, which is necessary before they are applied broadly [59,106].

And Finally, it is crucial to address the related ethical, medicolegal, and regulatory challenges as more AI technologies are developed that have the potential for clinical translation. There are a lot of unsolved questions on the ethical front. What situations must doctors tell their patients they're using AI techniques in their clinical workup? It might be crucial in

scenarios where AI functions as a "black box," in which clinicians act on the output of an AI tool without knowing how the algorithm came to its conclusion. When an AI technology misses a cancer, who is responsible? How much should be under human control? an the DL CAD systems make final decisions? Who is responsible for bad DL decisions? Will radiologists be biased as a result of AI assistance? What are people's perceptions of DL decision tools? Can DL CAD algorithms correctly describe their thought process? Before DL models can be widely used in actual clinical settings, it is evident that there must be discussion of these algorithmic biases, which also raise ethical issues [59,106].

## Conclusion

This overview gathered evidence from systematic reviews that evaluated the use of AI tools in the detection and diagnosis of malignant tumors based on radiographic images. The detection and diagnosis of malignant tumors with the help of AI seems to be feasible and accurate with the use of different technologies, such as CAD systems, machine learning algorithms and radiomic analysis when compared with the traditional model. ML algorithms performed better when compared to DL methods. However, these systems yielded better performance in some specific types of tumors such as cancer breast cancer, prostate cancer and thyroid nodules. Although there are limitations regarding the generalization for all types of cancer, these AI tools might aid professionals, serving as an auxiliary and teaching tool, especially for less trained professionals. Therefore, further standardized and longitudinal studies should be performed by using AI algorithms for detecting malignant lesions on different imaging modalities, by using larger datasets. These future perspectives will enable a better understanding of AI use in clinical oncologic practice.

## Supporting information

**S1 Checklist. PRISMA 2009 checklist.**
(DOCX)

**S1 Table. Database search strategy.**
(DOCX)

**S2 Table. Excluded articles and reasons for exclusion (n = 23).** Legend—**1**—Studies evaluating diagnosis of areas other than medicine and dentistry (Physiotherapist, Nutritionist, Nurse, Caregivers etc.); 2 –Patients with a confirmed diagnosis of cancer; 3—Systematic Reviews not evaluating the diagnostic accuracy Artificial intelligence, Machine learning, Deep learning and Convolutional Neural Networks; 4—Systematic Reviews with Artificial intelligence use for other diseases diagnosis (Diabetes, Hypertension, etc); 5—Systematic reviews in which AI was not compared to a reference test; 6—Systematic reviews evaluating other technologies for early detection or cancer diagnosis (spectrometry, biomarkers, autofluorescence, Multispectral widefield optical imaging, optical instruments, robotic equipment etc.); 7—literature reviews, integrative reviews, narrative reviews, overviews; 8—Editorials⁄Letters; 9—Conferences, Summaries, abstracts and posters; 10—In vitro studies; 11—Studies of animal models; 12—Thesis and Dissertations and book chapters; 13—Pipelines, guidelines and research protocols; 14—Review papers that do not follow the inclusion criteria adopted for the definition of Systematic Reviews; 15—Primary studies of any type; 16—No full paper avaliable.
(DOC)

**S3 Table. Overlaping (n = 09).**
(DOCX)

## Author Contributions

**Conceptualization:** Helbert Eustáquio Cardoso da Silva, Glaucia Nize Martins Santos, Cristine Miron Stefani, Nilce Santos de Melo.

**Data curation:** Helbert Eustáquio Cardoso da Silva, Glaucia Nize Martins Santos, André Ferreira Leite, Carla Ruffeil Moreira Mesquita, Paulo Tadeu de Souza Figueiredo.

**Formal analysis:** André Ferreira Leite, Carla Ruffeil Moreira Mesquita, Paulo Tadeu de Souza Figueiredo.

**Investigation:** André Ferreira Leite, Carla Ruffeil Moreira Mesquita, Paulo Tadeu de Souza Figueiredo.

**Methodology:** Helbert Eustáquio Cardoso da Silva, Glaucia Nize Martins Santos, Cristine Miron Stefani, Nilce Santos de Melo.

**Project administration:** Cristine Miron Stefani, Nilce Santos de Melo.

**Supervision:** Cristine Miron Stefani, Nilce Santos de Melo.

**Validation:** Cristine Miron Stefani, Nilce Santos de Melo.

**Writing – original draft:** Helbert Eustáquio Cardoso da Silva, Glaucia Nize Martins Santos, Cristine Miron Stefani, Nilce Santos de Melo.

**Writing – review & editing:** Helbert Eustáquio Cardoso da Silva, Glaucia Nize Martins Santos, André Ferreira Leite, Carla Ruffeil Moreira Mesquita, Paulo Tadeu de Souza Figueiredo, Cristine Miron Stefani, Nilce Santos de Melo.

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
