## [Decision Letter · Decision Letter 0]

17 Jan 2023

PONE-D-22-32314The use of artificial intelligence tools in cancer detection compared to the traditional diagnostic imaging methods: an overviewPLOS ONE

Dear Dr. SILVA,

Thank you for submitting your manuscript to PLOS ONE. After careful consideration, we feel that it has merit but does not fully meet PLOS ONE’s publication criteria as it currently stands. Therefore, we invite you to submit a revised version of the manuscript that addresses the points raised during the review process.

We look forward to receiving your revised manuscript.

Kind regards,

Azhar Imran, Ph.D

Academic Editor

PLOS ONE

Journal Requirements:

2. Our staff editors have determined that your manuscript is likely within the scope of our Early Detection, Screening and Diagnosis of Cancer Call for Papers. This editorial initiative is headed by in-house PLOS editors. This Call for Papers aims to explore recent advances in the early detection of cancer and implications of these advances for patient survival. Additional information can be found on our announcement page: https://collections.plos.org/call-for-papers/early-detection-screening-and-diagnosis-of-cancer/

If you would like your manuscript to be considered for this collection, please let us know in your cover letter and we will ensure that your paper is treated as if you were responding to this call. Please note that being considered for the Call for Papers does not require additional peer review beyond the journal’s standard process and will not delay the publication of your manuscript if it is accepted by PLOS ONE. If you would prefer to remove your manuscript from collection consideration, please specify this in the cover letter.

3. For clarity purposes please amend your title so that it better reflects the study type and findings with something like 'The use of artificial intelligence tools in cancer detection compared to the traditional diagnostic imaging methods: A Systematic Review'

Additional Editor Comments:

This paper reviews the application of different artificial intelligence (AI) technologies in cancer detection and then they compared them to the traditional diagnostic imaging methods. I think this is a very important and very big domain to be covered. According to comments from three reviews as well as the review by myself, a major revision for this manuscript is required. Please carefully addressed the concerns proposed by the reviewers, and prepare a point-to-point response as well.

Reviewers' comments:

Reviewer's Responses to Questions

**Comments to the Author**

1. Is the manuscript technically sound, and do the data support the conclusions?

Reviewer #1: Partly

Reviewer #2: Partly

2. Has the statistical analysis been performed appropriately and rigorously? 

Reviewer #1: No

Reviewer #2: N/A

3. Have the authors made all data underlying the findings in their manuscript fully available?

Reviewer #1: No

Reviewer #2: Yes

4. Is the manuscript presented in an intelligible fashion and written in standard English?

Reviewer #1: Yes

Reviewer #2: Yes

5. Review Comments to the Author

Reviewer #1: This paper reviews the application of different artificial intelligence (AI) technologies in cancer detection and then they compared them to the traditional diagnostic imaging methods. I think this is a very important and very big domain to be covered. However, I feel the paper has several serious issues which makes it difficult to follow it. So, I strongly suggest addressing the following points carefully:

1. How is this review paper different from other recently published articles (2021 and 2022)?

2. What do you mean by the traditional diagnostic imaging methods? Where did you provide the comparison?

3. There are lots of missed studies in cancer detection/prediction by ML/DL methods such as:

>>> A new nested ensemble technique for automated diagnosis of breast cancer

>>> A performance based study on deep learning algorithms in the effective prediction of breast cancer

>>> Pathologist-level interpretable whole-slide cancer diagnosis with deep learning

>>> High-accuracy prostate cancer pathology using deep learning

>>> CWV-BANN-SVM ensemble learning classifier for an accurate diagnosis of breast cancer

>>> Uncertainty quantification in skin cancer classification using three-way decision-based Bayesian deep learning

>>> MCUa: Multi-level context and uncertainty aware dynamic deep ensemble for breast cancer histology image classification

>>> Dermatologist-level classification of skin cancer with deep neural networks

>>> Survey on machine learning and deep learning applications in breast cancer diagnosis

>>> The application of deep learning in cancer prognosis prediction

>>> Personalized deep learning of individual immunopeptidomes to identify neoantigens for cancer vaccines

>>> A new era: artificial intelligence and machine learning in prostate cancer

>>> An enhanced deep learning approach for brain cancer MRI images classification using residual networks

Could the authors explain why these studies are not reviewed?

4. Please discuss the most important existing cancer databases.

5. As this paper aims to highlight the impact of Machine Learning (ML) and Deep Learning (DL) methods in cancer detection, it is highly recommended that the authors provide more results and discussions on the proposed ML/DL methods (theoretical) and not just providing their names in general.

6. Discussion: Which Machine Learning (ML) and Deep Learning (DL) methods are best in detecting cancers?

7. Please list and discuss the major limitations in the domain and literature.

8. The authors are highly encouraged to provide a new section including main research gaps and future ML/DL research directions.

9. The authors could discuss several important topics in the application of ML/DL methods in cancer detection. This means that discussions on uncertainty quantification and explainability (Explainable Artificial Intelligence (XAI)) of such methods should be considered. For more information the authors can use the following fully related studies:

>>> A review of uncertainty quantification in deep learning: Techniques, applications and challenges

>>> The need for quantification of uncertainty in artificial intelligence for clinical data analysis: increasing the level of trust in the decision-making process

>>> Quantifying uncertainty in machine learning classifiers for medical imaging

>>> XAI—Explainable artificial intelligence

>>> Second opinion needed: communicating uncertainty in medical machine learning

>>> Explainable Artificial Intelligence (XAI): Concepts, taxonomies, opportunities and challenges toward responsible AI

>>> Hercules: Deep Hierarchical Attentive Multilevel Fusion Model With Uncertainty Quantification for Medical Image Classification

>>> Explanation in artificial intelligence: Insights from the social sciences

>>> Explainable AI: Beware of inmates running the asylum or: How I learnt to stop worrying and love the social and behavioural sciences

Reviewer #2: The authors present their comprehensive overview of systematic reviews of applications of AI tools as applied to medical imaging of diagnosing malignant tumors in clinic, which is a very important topic. I have a few comments for the authors:

1. In this review, the three terms, AI, radiomics, and CAD were used interchangeably somehow. I would suggest the author to explain the definitions and relationships of these three terms in the Introduction section, and avoid to mix these concepts.

2. In the Conclusions of the Abstract section, it reads: " When compared to standard diagnostic imaging, AI applications for cancer detection may be more accurate in adult patients". I understand this point was the aim for this overview. However, in the discussion and conclusion section of this review, I could not find similar and supporting statements for this point. Please add more discussions on this.

3. As the authors stated in the manuscript, the meta-analysis of the data was unfeasible because of the high methodological heterogeneity of these secondary research, I would suggest the authors to consider to summarize the primary studies on this topic for a meta-analysis.

6. PLOS authors have the option to publish the peer review history of their article (what does this mean?). If published, this will include your full peer review and any attached files.

Reviewer #1: No

Reviewer #2: No

---

## [Author Response · Author response to Decision Letter 0]

28 Feb 2023

Azhar Imran, Ph.D Brasilia, February 28th, 2023

Academic Editor

PLOS ONE 

RE: Manuscript No. PONE-D-22-32314

Dear Dr. Azhar,

We would like to thank you for the opportunity to revise our manuscript entitled “The use of artificial intelligence tools in cancer detection compared to the traditional diagnostic imaging methods: an overview of systematic reviews”. We are also grateful for the valuable and insightful considerations provided by the reviewers. We are sure that the changes resulting from their comments improved the final quality of our manuscript. As the staff editors considered that our manuscript fits the scope of the Early Detection, Screening and Diagnosis of Cancer Call for Papers, we agreed with this statement and included it in the cover letter, as requested.

All suggestions were accepted, and further comments are given in detail. Text changes are highlighted in yellow in the manuscript. We listed the responses to all the reviewer’s comments, hoping that the revised manuscript will be considered for publication after the modifications. 

Reviewer #1:

1. Comment: How is this review paper different from other recently published articles (2021 and 2022)? 

Our response: The authors' intention was to carry out an overview that gathered the knowledge already published by systematic reviews in the literature regarding the use of artificial intelligence in the detection of cancer and its comparison with traditional imaging diagnostic methods in a single publication, aggregating the main evidence from articles of greater scientific robustness. A justification for carrying out the overview was added in the Introduction section on page 6 in lines 158-160.

2. Comment: What do you mean by the traditional diagnostic imaging methods? Where did you provide the comparison? 

Our response: It is the method that the radiologist/radiology resident uses, either by single reading or by double reading, to perform the evaluation of radiological images for cancer detection. The comparison can be found in the Session Discussion, page 24-25, lines 480-514.

3. Comment: There are lots of missed studies in cancer detection/prediction by ML/DL methods such as:...Could the authors explain why these studies are not reviewed? 

Our response: The overview carried out had as one of the inclusion criteria the use only of systematic review studies that addressed the relationship between the use of artificial intelligence tools in the detection of cancer and its comparison with traditional imaging diagnostic methods. This choice is due to the fact that the systematic review is a type of paper that is focused on a well-defined question, which aims to identify, select, evaluate and synthesize the relevant available evidence, with a critical analysis of the literature and a minimum of bias. The studies pointed out by Reviewer #1, despite addressing the subject of this overview, they are not systematic reviews to be selected according to the eligibility criteria listed in this overview. In addition, some studies used histological slides in the detection of cancer and the focus of the da overview was on the use of medical radiological images. Thus, they are not part of the selected studies. 

4. Comment: Please discuss the most important existing cancer databases.

Our response: The Discuss of the most important existing cancer databases has been added to the Discussion Session on page 30-31, lines 634-657, as suggested by Reviewer #1

5. Comment: As this paper aims to highlight the impact of Machine Learning (ML) and Deep Learning (DL) methods in cancer detection, it is highly recommended that the authors provide more results and discussions on the proposed ML/DL methods (theoretical) and not just providing their names in general. 

Our response: Discussion of the impact of Machine Learning (ML) and Deep Learning (DL) methods in cancer detection is described in the Discussion section on page 27-28, lines 564-600 as suggested by Reviewer #1

6. Comment: Discussion: Which Machine Learning (ML) and Deep Learning (DL) methods are best in detecting cancers? 

Our response: The authors agree with reviewer #1 and the discussion of Which Machine Learning (ML) and Deep Learning (DL) methods are best in detecting cancers is in the Discussion section, on page 25-26, lines 520-548. And in the Conclusion section page 38, lines 864.

7. Comment: Please list and discuss the major limitations in the domain and literature.

Our response: Authors agree with reviewer #1 and limitations related to selected systematic reviews and limitations related to overview are described on pages 35-36 lines 772-795.

8. Comment: The authors are highly encouraged to provide a new section including main research gaps and future ML/DL research directions.

Our response: The authors agree with Reviewer#1 and a new section including main research gaps and future ML/DL research directions added in Section Discussion on page 36-38 on lines 797-857.

9. Comment: The authors could discuss several important topics in the application of ML/DL methods in cancer detection. This means that discussions on uncertainty quantification and explainability (Explainable Artificial Intelligence (XAI)) of such methods should be considered. For more information the authors can use the following fully related studies:... 

Our response: The authors created two new subsections in the discussion section to discuss the two topics suggested by Reviewer #1. The subject of Explainable Artificial Intelligence (XAI) is in the subsection "The role of explainable artificial intelligence in DL and ML models" on page 31-33, lines 659-720. And the subject of uncertainty quantification is in the subsection "The use of uncertainty quantification approaches in medical imaging" on page 33-35, lines 722-770.

Reviewer #2:

1. Comment: In this review, the three terms, AI, radiomics, and CAD were used interchangeably somehow. I would suggest the author to explain the definitions and relationships of these three terms in the Introduction section, and avoid to mix these concepts. 

Our response: The authors agree with reviewer #2 and a further explanation of the definitions and relationships of these three terms has been placed in the Introduction section on page 5-6 at lines 131-157.

2. Comment: In the Conclusions of the Abstract section, it reads: " When compared to standard diagnostic imaging, AI applications for cancer detection may be more accurate in adult patients". I understand this point was the aim for this overview. However, in the discussion and conclusion section of this review, I could not find similar and supporting statements for this point. Please add more discussions on this. 

Our response: After reading the discussion session, the authors agreed with reviewer #2 and modified the conclusion of the abstract in order to make it more consistent with the final conclusions of the overview. Changes are on page 2-3, lines 49-55.

3. Comment: As the authors stated in the manuscript, the meta-analysis of the data was unfeasible because of the high methodological heterogeneity of these secondary research, I would suggest the authors to consider to summarize the primary studies on this topic for a meta-analysis. 

Our response: Although the Cochrane Manual for Systematic Reviews of Interventions considers the possibility of meta-analysis for overviews, the authors wishing to re-analyse outcome data may only be able to do so if the clinical parameters and statistics aspects of the included systematic reviews are sufficiently reported1. 

Due to the great heterogeneity of studies, data and types of cancer. In addition, three systematic reviews presented all the data, three systematic reviews presented partial data and three systematic reviews did not present enough data, even though the authors were requested by email and social networks, with no return response. Thus, the authors came to the conclusion that it would not be possible to carry out a meta-analysis of the primary studies of the selected systematic reviews. However, if the reviewers still deem it important and necessary to better qualify the knowledge presented in the overview, the authors ask for an extension of the deadline to carry out the meta-analysis of systematic reviews that present complete data.

1. Pollock M, Fernandes RM, Becker LA, Pieper D, Hartling L. Chapter V: Overviews of Reviews. In: Higgins JPT, Thomas J, Chandler J, Cumpston M, Li T, Page MJ, Welch VA (editors). Cochrane Handbook for Systematic Reviews of Interventions version 6.3 (updated February 2022). Cochrane, 2022. Available from www.training.cochrane.org/handbook.

We remain at your disposal for any further clarification that you may require regarding our manuscript. We forward to your reply. Thank you for your consideration.

Sincerely,

Helbert Eustáquio Cardoso da Silva, in behalf of the all authors

University of Brasilia, Health Sciences Faculty, Brasília, DF, Brazil. 

e-mail: helbertcardososilva@gmail.com

---

## [Decision Letter · Decision Letter 1]

30 May 2023

PONE-D-22-32314R1The use of artificial intelligence tools in cancer detection compared to the traditional diagnostic imaging methods: an overview of the systematic reviewsPLOS ONE

Dear Dr. SILVA,

Thank you for submitting your manuscript to PLOS ONE. After careful consideration, we feel that it has merit but does not fully meet PLOS ONE’s publication criteria as it currently stands. Therefore, we invite you to submit a revised version of the manuscript that addresses the points raised during the review process.

We look forward to receiving your revised manuscript.

Kind regards,

Yuchen Qiu, Ph.D.

Academic Editor

PLOS ONE

Reviewers' comments:

Reviewer's Responses to Questions

**Comments to the Author**

1. If the authors have adequately addressed your comments raised in a previous round of review and you feel that this manuscript is now acceptable for publication, you may indicate that here to bypass the “Comments to the Author” section, enter your conflict of interest statement in the “Confidential to Editor” section, and submit your "Accept" recommendation.

Reviewer #1: All comments have been addressed

Reviewer #2: (No Response)

2. Is the manuscript technically sound, and do the data support the conclusions?

Reviewer #1: (No Response)

Reviewer #2: (No Response)

3. Has the statistical analysis been performed appropriately and rigorously? 

Reviewer #1: (No Response)

Reviewer #2: (No Response)

4. Have the authors made all data underlying the findings in their manuscript fully available?

Reviewer #1: (No Response)

Reviewer #2: (No Response)

5. Is the manuscript presented in an intelligible fashion and written in standard English?

Reviewer #1: (No Response)

Reviewer #2: (No Response)

6. Review Comments to the Author

Reviewer #1: In my view, the authors have improved the paper considerably and have addressed all my previous concerns carefully. I therefore think that the paper can be accepted.

Reviewer #2: As suggested by my previous comment 3, I think the authors should take more time to carry out the meta-analysis of systematic reviews that present complete data on this important topic.

7. PLOS authors have the option to publish the peer review history of their article (what does this mean?). If published, this will include your full peer review and any attached files.

Reviewer #1: No

Reviewer #2: No

---

## [Author Response · Author response to Decision Letter 1]

22 Jun 2023

Azhar Imran, Ph.D Brasilia, June 22th, 2023

Academic Editor

PLOS ONE 

RE: Manuscript No. PONE-D-22-32314

Dear Dr. Azhar,

We would like to thank you for the opportunity to revise our manuscript entitled “The use of artificial intelligence tools in cancer detection compared to the traditional diagnostic imaging methods: an overview of systematic reviews”. We are also grateful for the valuable and insightful considerations provided by the reviewers. We are sure that the changes resulting from their comments improved the final quality of our manuscript. As the staff editors considered that our manuscript fits the scope of the Early Detection, Screening, and Diagnosis of Cancer Call for Papers, we agreed with this statement and included it in the cover letter, as requested.

All suggestions were accepted, and further comments are given in detail. Text changes are highlighted in yellow in the manuscript. We listed the responses to all the reviewer’s comments, hoping that the revised manuscript will be considered for publication after the modifications. 

Corrections by authors:

 The authors, after another careful reading, made some changes to the text of the manuscript, which will be highlighted below, with the aim of improving the understanding of the text, without changing the structure already approved by the reviewers.

 The Background and purpose in the Abstract section, lines 28 to 30, has been replaced by the following sentence: "In comparison to conventional medical imaging diagnostic modalities, the aim of this overview article is to analyze the accuracy of the application of artificial intelligence (AI) techniques in the identification and diagnosis of malignant tumors in adult patients."

 The expression "are more likely" was replaced by "are prone" on line 113.

 The expression "systematic review" has been replaced by "overview" on line 172.

 The following sentence was added in the Materials and methods section, in lines 174 to 176: "and was developed according to the JBI Manual for Evidence Synthesis (https://synthesismanual.jbi.global) and the Cochrane Handbook for Systematic Reviews (www .training.cochrane.org/handbook)."

 The sentence was rewritten in the section Materials and methods, in lines 177 to 179: "The definition of systematic reviews considered was that established by the Cochrane Collaboration. A study was considered a systematic review when reporting or including:"

 The sentence was rewritten in the Materials and methods section, in lines 258 to 262: " It should be noted that critical appraisal/risk of bias tools classically indicated for systematic reviews, such as AMSTAR 2 and ROBIS, were designed for systematic reviews of intervention, while the articles included were systematic reviews of diagnostic accuracy. We opted for performing the methodological assessment, not the risk of bias in the selected studies."

 In the Results section, the initials RS were replaced by the initials SR due to a spelling error in lines 284, 285 and 286.

 The expression "quality" was replaced by "certainty" on line 446.

 In the Discussion section, the abbreviation ML was replaced by the expression "Machine learning" for a better understanding of the sentence, on line 570.

 In the Discussion section, on line 850, the term "possible" was added in the sentence "possible presence of publication bias due to lack of studies with unfavorable data" for a better understanding of the sentence.

Reviewer #2:

1. Comment: As suggested by my previous comment 3, I think the authors should take more time to carry out the meta-analysis of systematic reviews that present complete data on this important topic. 

Our response: We appreciate the suggestion and have considered it carefully. However, neither the Cochrane Handbook for Systematic Reviews1 nor the JBI Manual for Evidence Synthesis2 (used as the main guidance for conducting this overview) foreseen a quantitative synthesis for Overviews/Umbrella Reviews. Both suggest only a critical evaluation of the meta-analyses presented by the systematic reviews included in the overview.

In addition, the studied meta-analyses presented some limitations. First, all showed considerable heterogeneity with regard to sensitivity and specificity across studies. It is likely that this heterogeneity is related to methodological aspects of the different studies, and to the basic differences between the included patients. Second, the sample size of the included studies was relatively small (Dorrius 2011 and Zhao 2019, Eadie 2012). The authors considered the occurrence of selection (Dorrius 2011 and Zhao 2019, Eadie 2012), measurement (Zhao 2019) and publication (Zhao 2019, Eadie 2012) bias in performing the meta-analyses.

As a solution discussed by the authors, we decided to carry out a critical analysis of the only three studies that presented complete data (Dorrius et al., 2011, Zhao et al., 2019, Eadie et al., 2012), included in the Discussion section, lines 664 to 726, as a way of giving the closest answer to the reviewer's #2 request and consistent with what the literature recommends regarding performing an overview.

1Higgins JPT, Thomas J, Chandler J, Cumpston M, Li T, Page MJ, Welch VA (editors). Cochrane Handbook for Systematic Reviews of Interventions version 6.3 (updated February 2022). Cochrane, 2022. Available from www.training.cochrane.org/handbook. 

2Aromataris E, Munn Z (Editors). JBI Manual for Evidence Synthesis. JBI, 2020. Available from https://synthesismanual.jbi.global. https://doi.org/10.46658/JBIMES-20-01

We remain at your disposal for any further clarification that you may require regarding our manuscript. We look forward to your reply. Thank you for your consideration.

Sincerely,

Helbert Eustáquio Cardoso da Silva, on behalf of the authors

University of Brasilia, Health Sciences Faculty, Brasília, DF, Brazil. 

e-mail: helbertcardososilva@gmail.com

---

## [Decision Letter · Decision Letter 2]

12 Sep 2023

The use of artificial intelligence tools in cancer detection compared to the traditional diagnostic imaging methods: an overview of the systematic reviews

PONE-D-22-32314R2

Dear Dr. SILVA,

We’re pleased to inform you that your manuscript has been judged scientifically suitable for publication and will be formally accepted for publication once it meets all outstanding technical requirements.

Kind regards,

Yuchen Qiu, Ph.D.

Academic Editor

PLOS ONE

Additional Editor Comments (optional):

Reviewers' comments:

Reviewer's Responses to Questions

**Comments to the Author**

1. If the authors have adequately addressed your comments raised in a previous round of review and you feel that this manuscript is now acceptable for publication, you may indicate that here to bypass the “Comments to the Author” section, enter your conflict of interest statement in the “Confidential to Editor” section, and submit your "Accept" recommendation.

Reviewer #2: All comments have been addressed

2. Is the manuscript technically sound, and do the data support the conclusions?

Reviewer #2: Yes

3. Has the statistical analysis been performed appropriately and rigorously? 

Reviewer #2: N/A

4. Have the authors made all data underlying the findings in their manuscript fully available?

Reviewer #2: Yes

5. Is the manuscript presented in an intelligible fashion and written in standard English?

Reviewer #2: Yes

6. Review Comments to the Author

Reviewer #2: (No Response)

7. PLOS authors have the option to publish the peer review history of their article (what does this mean?). If published, this will include your full peer review and any attached files.

Reviewer #2: No

---

## [Editor Report · Acceptance letter]

25 Sep 2023

PONE-D-22-32314R2 

The use of artificial intelligence tools in cancer detection compared to the traditional diagnostic imaging methods: an overview of the systematic reviews 

Dear Dr. Silva:

I'm pleased to inform you that your manuscript has been deemed suitable for publication in PLOS ONE. Congratulations! Your manuscript is now with our production department. 

Kind regards, 

on behalf of

Dr. Yuchen Qiu 

Academic Editor

PLOS ONE